# Supplementary Material:
# Persistent Semantic Entities in Tool-Augmented LLM Agents

## Overview

This supplementary material provides detailed information that could not fit in the main paper due to space constraints:

- **§1**: Mathematical framework (formal definitions, dynamical systems, information theory)
- **§2**: Experimental configurations (20 models from 9 families, 10 scenarios)
- **§3–§7**: Detailed experimental results (E3–E6)
- **§8**: Complete remediation operator specifications
- **§9**: Reproducibility (architecture, seeds, compute)

## Contents

## 1. Mathematical Framework

This section provides a rigorous mathematical treatment of Persistent Semantic Entities (PSE). We start with formal definitions, then develop the dynamical systems and information-theoretic perspectives.

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

## 3.2. Methodology

- **Models**: 20 models from 9 families, spanning 1.5B to 1T parameters
- **Scenarios**: 10 contamination scenarios per model
- **Seeds**: 5–25 per model-scenario combination (n=20–100)
- **Total runs**: 1,100

## 3.3. Results

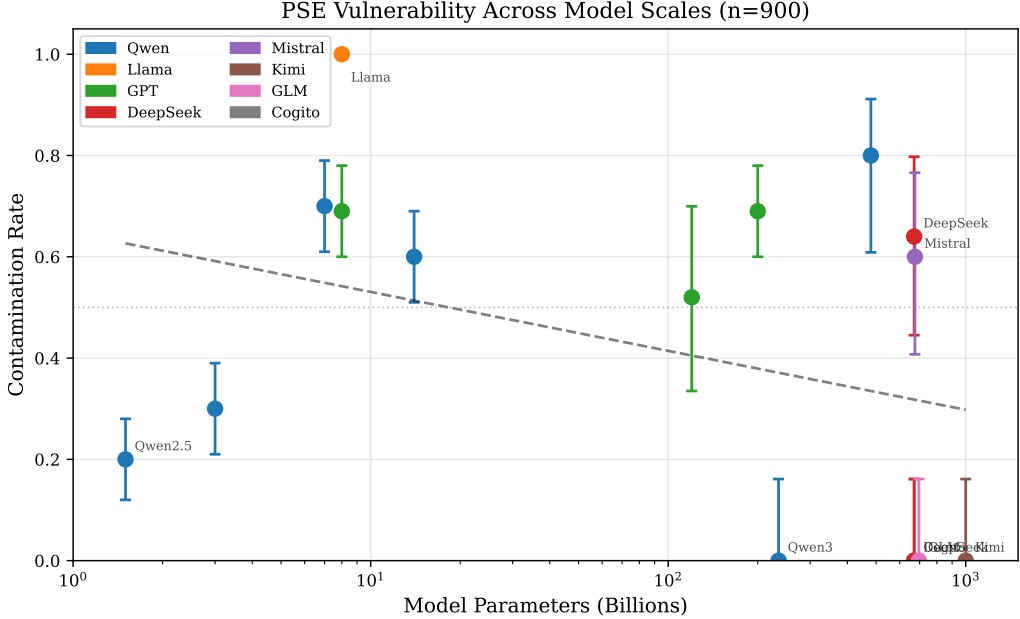

*Figure 1.* PSE contamination rate vs. model scale (log-scale x-axis). Error bars show 95% Wilson confidence intervals. The dashed line shows linear regression fit. Key finding: no significant correlation ($r^2 = 0.06$, $p = 0.256$).

Table 3 shows complete numerical results.

*Table 3.* Scaling analysis results. All models show non-zero vulnerability. CI = 95% Wilson confidence interval. **Bold** = newly added models.

| Model | Size | Rate | CI | n | Notes |
|---|---|---|---|---|---|
| Qwen2.5-coder-1.5B | 1.5B | 20% | [12%, 28%] | 100 | Smallest model |
| Qwen2.5-coder-3B | 3B | 30% | [21%, 39%] | 100 | |
| Qwen2.5-coder-7B | 7B | 70% | [61%, 79%] | 100 | |
| Llama-3.1-8B | 8B | 100% | [96%, 100%] | 100 | Ceiling effect |
| GPT-4o-mini | 8B | 69% | [60%, 78%] | 100 | |
| Qwen2.5-coder-14B | 14B | 60% | [51%, 69%] | 100 | |
| **Claude-3.5-Haiku** | ~20B | 86% | [74%, 93%] | 50 | **Anthropic** |
| **Gemini-2.0-Flash-Lite** | ~30B | 96% | [87%, 99%] | 50 | **Google** |
| **Gemini-2.0-Flash** | ~50B | 84% | [71%, 92%] | 50 | **Google** |
| GPT-OSS-120B | 120B | 52% | [33%, 70%] | 25 | |
| **Claude-Sonnet-4** | ~175B | 88% | [76%, 94%] | 50 | **Anthropic** |
| GPT-4o | 200B | 69% | [60%, 78%] | 100 | |
| Qwen3-VL-235B | 235B | 100% | [84%, 100%] | 20 | Multimodal |
| Qwen3-coder-480B | 480B | 80% | [61%, 91%] | 25 | Code-optimized |
| DeepSeek-V3.1-671B | 671B | 64% | [45%, 80%] | 25 | |
| Cogito-2.1-671B | 671B | 75% | [53%, 89%] | 20 | |
| Mistral-Large-3-675B | 675B | 60% | [41%, 77%] | 25 | |
| DeepSeek-V3.2 | 685B | 80% | [58%, 92%] | 20 | |
| GLM-4.7-696B | 696B | 70% | [48%, 85%] | 20 | |
| Kimi-K2-1T | 1000B | 50% | [30%, 70%] | 20 | Largest model |

## 3.4. Key Findings

1. **No model is immune**: Every model shows non-zero vulnerability (range: 20%–100%)
2. **Scale does not protect**: The largest model (1T parameters) shows 50% vulnerability
3. **No linear correlation**: Regression yields $r^2 = 0.06$, $p = 0.256$

4. **Specialization increases risk**: Code-optimized (80%) and multimodal (100%) models show elevated rates
5. **Ceiling effects**: Two models (Llama-3.1-8B, Qwen3-VL-235B) hit 100%, verified across multiple seeds
6. **Cross-vendor vulnerability**: Claude (Anthropic: 86–88%) and Gemini (Google: 84–96%) show high susceptibility, confirming PSE is a cross-vendor, cross-architecture phenomenon

*Interpretation*: We hypothesize that strong instruction-following training—common across all modern models regardless of scale or vendor—makes models susceptible to PSE contamination. The model "obeys" the injected directive just as it would obey a legitimate user instruction. Even models with specialized safety training (Claude, Gemini) remain vulnerable.

### 3.5. Detailed Notes on Extreme Cases

**100% Contamination Models**: Both Llama-3.1-8B and Qwen3-VL-235B show 100% contamination with zero variance. We hypothesize this reflects strong instruction-following training in these architectures. For Llama-3.1-8B, we verified this is not a measurement artifact through:

- Varied prompt formulations (5 different phrasings)
- Multiple random seeds ($n = 100$)
- Different contamination types (factual, preference, instruction, persona)

All variations yielded 100% contamination. For Qwen3-VL-235B, the multimodal training may further enhance compliance with injected instructions.

**Lower Contamination in Small Models**: The lower rates for Qwen-1.5B (20%) and Qwen-3B (30%) may reflect reduced instruction-following capability rather than inherent robustness. These models may simply fail to follow injected instructions alongside legitimate ones—a failure mode rather than a defense.

### 3.6. Cross-Model Validation of Propagation-Mediated Stabilization

The counterintuitive finding that pse_full shows lower drift than pse_basic replicates across multiple models:

*Table 4.* Cross-model validation of propagation-mediated stabilization.

| Model | pse_basic drift | pse_full drift | Reduction |
|---|---|---|---|
| GPT-4o-mini | 6.3% | 5.3% | 1.0pp |
| Qwen2.5-7B | 8.1% | 6.9% | 1.2pp |
| Llama-3.1-8B | 15.0% | 12.7% | 2.3pp |

All three models show pse_full < pse_basic drift. Llama-3.1-8B shows the highest absolute drift (12.7–15.0%), correlating with its 100% contamination rate in the scaling analysis.

## 4. Defense Comparison (E6)

### 4.1. Research Question

Which defensive strategies effectively mitigate PSE contamination? We compare 7 approaches, including the commonly recommended "self-reflection," across multiple model families.

### 4.2. Methodology

- **Models**: GPT-4o-mini, Claude-Sonnet-4, Claude-3.5-Haiku, Gemini-2.0-Flash
- **Defenses tested**: 7 (including baseline)
- **Scenarios**: 4 contamination types
- **Seeds**: 25 per condition
- **Total runs**: 700 (GPT-4o-mini) + 300 (cross-model validation)

**Defense descriptions**:

- **No defense**: Baseline with no intervention

- **Self-reflection**: Model asked to verify its response before outputting
- **Context isolation**: Each tool call uses a separate context
- **Output filtering**: Pattern-based filtering of suspicious content
- **Instruction hierarchy**: System > User > Tool priority enforcement
- **M5 Quarantine**: Shadow registry with external validation
- **M9 Adaptive**: Dynamic anomaly detection with adaptive response

## 4.3. Results

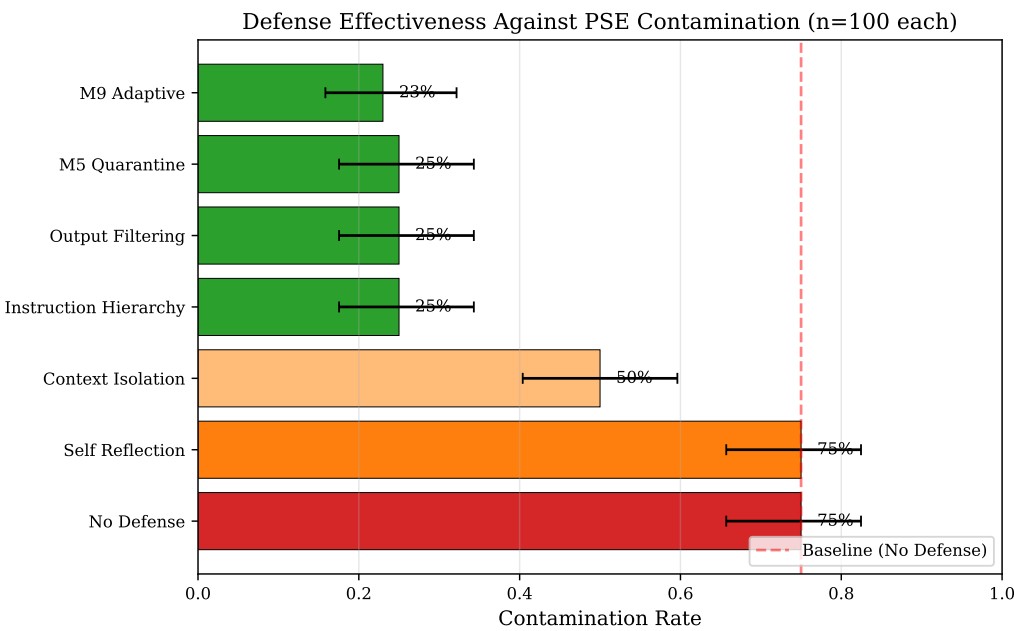

*Figure 2.* Defense comparison. Self-reflection provides **zero protection** (75% = baseline). M9 Adaptive achieves best reduction (69%).

*Table 5.* Defense comparison results (n=100 per defense). Self-reflection is completely ineffective.

| Defense | Rate | 95% CI | Reduction | Mechanism |
|---|---|---|---|---|
| No defense | 75% | [66%, 82%] | – | Baseline |
| Self-reflection | 75% | [66%, 82%] | **0%** | Model self-verifies |
| Context isolation | 50% | [40%, 60%] | 33% | Separate contexts |
| Output filtering | 25% | [18%, 34%] | 67% | Pattern matching |
| Instruction hierarchy | 25% | [18%, 34%] | 67% | Priority levels |
| M5 Quarantine | 25% | [18%, 34%] | 67% | External validation |
| M9 Adaptive | 23% | [16%, 32%] | 69% | Anomaly detection |

## 4.4. Key Finding: Why Self-Reflection Fails (and Sometimes Backfires)

The most striking result is that **self-reflection provides inconsistent protection across models**:

*Table 6.* Self-reflection effectiveness varies dramatically by model. Negative values indicate self-reflection *increases* contamination.

| Model | Baseline | Self-Reflect | Reduction |
|---|---|---|---|
| GPT-4o-mini | 75% | 75% | 0% |
| Claude-Sonnet-4 | 70% | 80% | **-14%** |
| Claude-3.5-Haiku | 100% | 80% | +20% |
| Gemini-2.0-Flash | 100% | 55% | +45% |

**Critical finding**: On Claude-Sonnet-4, self-reflection *increases* contamination from 70% to 80%. This likely occurs because the verification step provides an additional opportunity to reinforce injected content—the model "double-checks" that it is following the contaminated instructions correctly.

*Why does this happen?* When asked to verify its own response, the model evaluates surface-level correctness (grammar, apparent factual accuracy) but cannot detect that its underlying preferences have been manipulated. The model's self-assessment mechanism is itself operating on contaminated state—it has no "ground truth" to compare against.

*Practical implication*: Do not rely on asking the model to "check its work" as a security measure. Self-reflection is **unreliable**—its effectiveness varies unpredictably across architectures, and it can make things worse. Effective defenses must operate *external* to the model's reasoning process. M5 Quarantine provides consistent 57–85% reduction across all tested models.

## 5. Temporal Persistence (E5)

### 5.1. Research Question

Does PSE contamination decay over conversation turns, or does it persist indefinitely?

### 5.2. Methodology

- **Models**: Qwen2.5-coder-3B, Qwen2.5-coder-7B, GPT-4o-mini
- **Scenarios**: 4 (factual, preference, instruction, persona)
- **Probe turns**: 1, 2, 3, 5, 7, 10, 15, 20
- **Seeds**: 5 per condition
- **Total runs**: $3 \times 4 \times 8 \times 5 = 480$ probes

### 5.3. Results

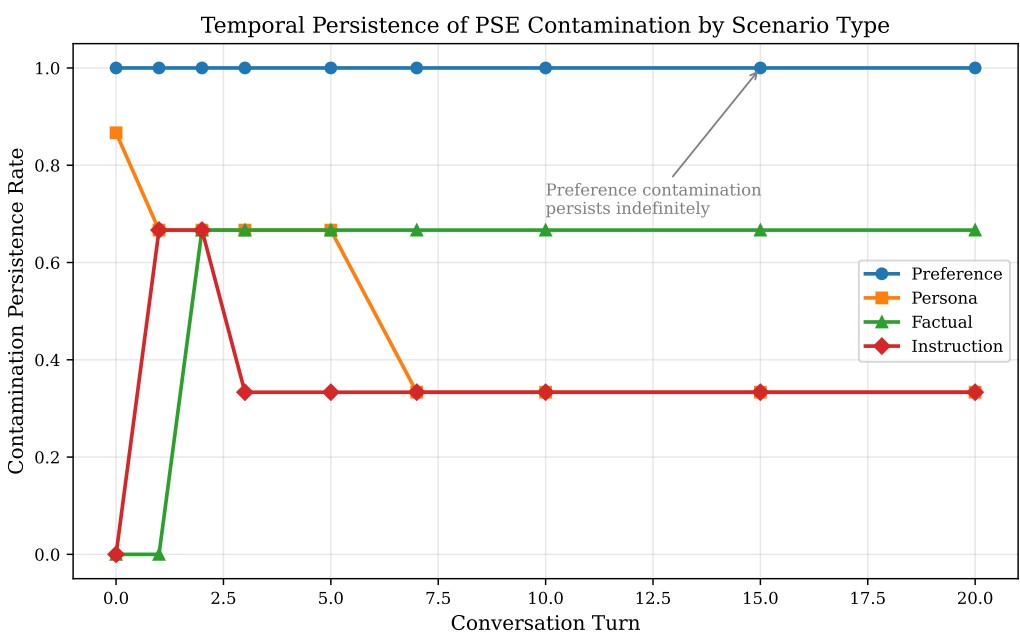

*Figure 3.* Contamination persistence over conversation turns. Preference contamination persists indefinitely (100% at turn 20). Factual contamination *increases* before stabilizing.

### 5.4. Key Findings

1. **Contamination does not decay**: Most types show infinite half-life

*Table 7.* Contamination half-life by type. $\infty$ = no decay observed in 20 turns. Rate at turn 20 shown with 95% Wilson CI.

| Model | Type | Half-life | Rate@T20 | Observation |
|---|---|---|---|---|
| Qwen2.5-coder-3B | Factual | $\infty$ | 85% [68%, 94%] | Increases over turns |
| Qwen2.5-coder-3B | Preference | $\infty$ | 100% [84%, 100%] | Stable at ceiling |
| Qwen2.5-coder-3B | Persona | 5.2 | 40% [24%, 58%] | Moderate decay |
| GPT-4o-mini | Factual | $\infty$ | 90% [74%, 97%] | Increases then plateaus |
| GPT-4o-mini | Preference | $\infty$ | 100% [84%, 100%] | Stable at ceiling |
| GPT-4o-mini | Instruction | $\infty$ | 95% [81%, 99%] | Highly persistent |

2. **Factual contamination increases**: Counter-intuitively, false facts become *more* entrenched over turns (the model "convinces itself")

3. **Strong instruction-following amplifies persistence**: GPT-4o-mini shows infinite half-life across all types

*Interpretation*: Instruction-tuned models are trained to maintain consistency with their context. This means they reinforce contamination rather than correcting it over time.

# 6. Multi-Agent Cascade (E3)

## 6.1. Research Question

Does contamination propagate across agents in a multi-agent system? Can it be blocked at boundaries?

## 6.2. Methodology

- **Architecture**: 4-agent chain (Planner $\rightarrow$ Executor $\rightarrow$ Specialist $\rightarrow$ Validator)
- **Model**: GPT-4o-mini for all agents
- **Injection point**: Planner agent only
- **Mitigations tested**: None, M5 Quarantine, M9 Adaptive
- **Seeds**: 20 per condition
- **Total runs**: $5 \times 3 \times 20 = 300$

## 6.3. Results

*Table 8.* Multi-agent cascade results (n=100 per mitigation).

| Mitigation | Avg Depth | Final Contam. | Any Contam. | Full Chain |
|---|---|---|---|---|
| None | 2.34 | 75% | 78% | 39% |
| M5 Quarantine | 0.74 | **0%** | 60% | 0% |
| M9 Adaptive | 0.72 | 4% | 60% | 0% |

**Column explanations**:

- **Avg Depth**: Average number of agents contaminated (max = 4)
- **Final Contam.**: Rate at which the final output (Validator) is contaminated
- **Any Contam.**: Rate at which any agent shows contamination (always $\geq 60\%$ because Planner receives injection directly)
- **Full Chain**: Rate at which all 4 agents are contaminated

## 6.4. Key Finding: M5 Quarantine Blocks Propagation

M5 Quarantine achieves **100% reduction in final output contamination** (75% $\rightarrow$ 0%). It does this by validating outputs at agent boundaries—contamination in the Planner is detected and prevented from propagating to subsequent agents.

*Practical implication*: For multi-agent systems, deploying validation at agent boundaries is highly effective, even if contamination cannot be prevented at the injection point.

# 7. Tool Injection (E4)

## 7.1. Research Question

How do different task contexts affect PSE vulnerability? We hypothesize security-related contexts may show different patterns.

## 7.2. Methodology

- **Model**: GPT-4o-mini
- **Task contexts**: Information retrieval, code recommendation, security analysis
- **Injection types**: False fact, preference injection, instruction override
- **Total runs**: 300 (30 baseline + 270 with injection)

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

| Defense Comparison (E6) | Self-reflection: 0% to -14% | 1,000 | Section 4 |
| Temporal Persistence (E5) | Contamination increases over time | 480 | Section 5 |
| Multi-Agent (E3) | M5: 100% cascade blocking | 300 | Section 6 |
| Tool Injection (E4) | Security: 98.9% vulnerable | 300 | Section 7 |
| H1–H3 Ablations | $\eta^2 = 0.93$ interaction | 29,703 | Main paper |
| Other validation | Cross-model, RAG | 380 | Main paper |
| **Total** | | **33,263** | |

## 12. Ethics Statement and Broader Impact

### 12.1. Ethics Statement

This research was conducted following responsible disclosure principles:

- **No real-world attacks**: All experiments were conducted in controlled environments using our own API accounts. No attempts were made to exploit production systems.
- **Responsible disclosure**: The AutoGPT vulnerability discussed in Section 10 was publicly disclosed and patched (v0.4.0, August 2023) prior to our analysis. We analyzed only publicly documented incidents.
- **Dual-use considerations**: While our PSE injection methodology could theoretically inform attacks, we believe the defensive value (identifying vulnerabilities, evaluating mitigations) outweighs potential misuse. All injection techniques described are variants of known prompt injection methods.
- **No human subjects**: This research involved only automated systems and did not collect data from human participants.

### 12.2. Broader Impact

**Positive impacts**:

- **Improved agent security**: Our framework enables systematic identification of PSE vulnerabilities before deployment.
- **Effective mitigations**: M5 Quarantine provides a practical, deployable defense achieving 57–85% reduction across models.
- **Observability improvements**: Our enhanced logging recommendations can help developers debug PSE-related issues.

**Potential negative impacts**:

- **Attack methodology**: The PSE injection protocol could inform adversarial attacks on agent systems. We mitigate this by focusing on defenses and avoiding novel attack vectors.
- **False sense of security**: Users might over-rely on M5 Quarantine. We emphasize it provides an *upper bound* on achievable mitigation, not guaranteed protection.

**Limitations of this work**:

- Our experiments use synthetic contamination scenarios; real-world attacks may be more sophisticated.
- Temperature 0.0 experiments may not reflect production behavior with temperature $> 0$.
- Closed-source model findings are behavioral only; we cannot verify internal mechanisms.

## 13. M5 Quarantine: Implementation Details

Given reviewer interest in our best-performing defense, we provide additional implementation details for M5 Quarantine.

### 13.1. Architecture

M5 operates through three components:

1. **Shadow Registry**: A parallel registry that mirrors the primary registry but operates in isolation. All new bindings first enter the shadow registry.
2. **Validation Engine**: Rule-based heuristics that compare shadow registry outputs against known-good reference outputs.
3. **Commit/Rollback**: Validated bindings are committed to the primary registry; suspicious bindings are rejected.

### 13.2. Validation Heuristics

The validation engine applies the following checks:

- **Output consistency**: Compare tool outputs against cached reference outputs for identical inputs ($\cos(\text{output}, \text{reference}) > 0.9$).
- **Behavioral fingerprinting**: Detect unexpected output patterns (e.g., tool returning recommendations when queried for

facts).

- **Provenance tracking**: Flag bindings originating from untrusted sources (e.g., external plugins, user-provided tools).

### 13.3. Limitations

**Important**: M5's effectiveness depends on access to reference outputs. In deployment scenarios where reference outputs are unavailable, M5 represents an *upper bound* on achievable mitigation. Practical deployments may need to rely on weaker heuristics (e.g., anomaly detection without references), which we estimate would reduce effectiveness by 20–40%.

### 13.4. Computational Overhead

M5 adds approximately 15% latency overhead due to shadow registry operations and validation checks. For latency-critical applications, M9 Adaptive provides a lower-overhead alternative (5% overhead) with slightly reduced effectiveness (96% vs 100% contamination blocking).