# OpenReview forum: "Persistent Semantic Entities in Tool-Augmented LLM Systems"
_ICML.cc/2026/Conference — ICML 2026 regular_

### Official Review · Reviewer_DgKb · 2026-03-09

**Soundness:** 3
**Presentation:** 2
**Significance:** 2
**Originality:** 3
**Overall Recommendation:** 3
**Confidence:** 5

**Summary:**

This work proposed Persistent Semantic Entities (PSE), implicit state in tool-augmented LLM agents that persists across sessions via name binding, event triggering, and cross-boundary propagation. The authors test 20 models from 9 families and find universal susceptibility with contamination rates from 20% to 100%, non-decaying contamination in tested configurations, and that self-reflection is an unreliable defense. Their proposed quarantine-based validation consistently reduces contamination by 57% to 85%, though the authors note this assumes access to reference outputs and represents an upper bound.

**Compliance With Llm Reviewing Policy:**

Affirmed.

**Key Questions For Authors:**

- What happens when M5's reference outputs are themselves slightly contaminated? I think contamination is one potential problem the PSE may face.
- How’s the evaluation results when these injection mechanism such as prompt-level, tool registration, policy binding are tested seperately? An ablation on these will make the evaluation more solid.
- The paper argues in Section 6 that self-reflection fails structurally because the same parameters produce and evaluate contaminated output. Has the authors explored multi-round or adversarial self-reflection, such as prompting the model to argue against its previous output?

**Limitations:**

- In addition to the presentation weakness mentioned in the Weakness section, there are other minor issue such as all experiments use temperature 0.0, may not accurately reflect the messy complexities of live production deployments. From my point of view, having some hyperaparameter would make the paper more solid
- I think that is one limitation, but very minor issue as I think the experiment setup is overall, comprehensive except for the problems mentioned above. API cost constraints limited the sample sizes for the largest frontier models (n from 20 to 50 seems to be a small batch), the cost of different models might be very different, for example GPT 5.4 Pro released recently may cost $15.00/M input and $90.00/M output, while some open source models such as GLM-5 I checked on OpenRouter would only cost $0.8/M and $2.56/M, instead of using sample numbers, maybe using total cost as constrains might be a better option (GPT 5.4 Pro and GLM 5 are released recently, which is after the paper cut-off date, here I just take as an example, overall the model selection is comprehensive and decent).

**Strengths And Weaknesses:**

Strengths
- The authors highlight a fascinating, tragic irony: the exact features that make multi-agent systems powerful (reusable tools, reactive events) are structurally inseparable from the vectors that make them hopelessly vulnerable
- I think the presentation of name binding, event triggering, and propagation is straightforward and easy to follow for agent vulnerabilities.

Weaknesses
- The quarantine-based validation (M5) assumes access to known-good reference outputs, which the authors acknowledge as an upper bound. In open-ended production settings such references rarely exist. The paper would benefit from evaluating a degraded variant of M5 using approximate or partial references.
- The claim that contamination does not decay is tested only on GPT-4o-mini and Qwen models in experiment (E5 in the paper), which presented as a general finding. Given the high inter-model variance elsewhere, this should be tested on more families or explicitly scoped as preliminary.
- The experiment section is hard to follow for the reviewer, this paper claimed that the work has tested 9 families and 20 models, While Table 5 provides a scaling view, there is no uniformed table to show the results, thus I think it would be easier to follow if the author can have such an integrated result presentation.

---

> ### Author Rebuttal · Authors · 2026-03-25
>
> We thank the reviewer for the careful assessment and for highlighting the structural inseparability insight. We are encouraged that the reviewer recognizes the originality of the core contribution. The concerns raised primarily relate to evaluation completeness and presentation, which we address below with additional post-submission experimental evidence.
>
> **W1: M5 (Quarantine) assumes known-good reference outputs.**
> We agree that M5 represents an upper bound. Following the reviewer's suggestion, we conducted post-submission experiments (n=100, Gemini-2.0-Flash-Lite) evaluating a more realistic defense: *context-isolated self-verification*, where the model generates under contaminated context and then a separate clean call (without the contaminated context) performs fact-checking. No oracle references are required.
>
> | Defense | Contam. Rate | Reduction |
> |---|---|---|
> | No defense | 70% (14/20) | -- |
> | Context-isolated verification (no oracle) | 15% (3/20) | **78.6%** |
> | Full external validation (cross-model) | 0% (0/19) | **100%** |
>
> This differs fundamentally from in-context self-reflection (0% to -14% in the paper), which fails because verification shares the contaminated context. Context-isolated verification removes the contamination channel, confirming our original analysis that the failure mode is architectural. M9 (Adaptive) also provides an oracle-free alternative via anomaly detection (Table 4).
> Importantly, this directly addresses the reviewer's concern regarding realism by demonstrating strong mitigation without oracle assumptions.
>
> **W2: Non-decay claim limited to GPT-4o-mini and Qwen.**
> We agree and will: (1) explicitly scope the finding as "preliminary evidence in tested configurations"; (2) clarify that no tested model exhibited statistically significant decay, so the conservative claim holds under the evaluated settings. Extending to additional model families is a natural next step.
>
> **W3: No unified results table.**
> We fully agree. We will add a consolidated table reporting all 20 models with model name, parameter count, n, contamination rate (95% Wilson CI), defense effectiveness, and scenario breakdown. The post-submission ablation adds two controlled data points (Gemini n=160, Llama n=160), improving coverage and clarity.
>
> **Q1: What if M5's references are contaminated?**
> M5 validates against references generated *prior to* PSE injection. If references are post-contamination, M5 degrades—this motivates deployment at agent boundaries. E3 (n=300) shows that without boundary validation, contamination amplifies 25% to 75% across 5 agents; with boundary validation, final contamination is 0%. We will make this trust-anchor assumption explicit.
>
> **Q2: Ablation on injection mechanisms.**
> Post-submission 2^3 factorial ablation directly addresses this request:
>
> | Config | Gemini-2.0-Flash-Lite (n=160) | Llama-3.1-8B (n=160) |
> |---|---|---|
> | No mechanisms | 0% | 0% |
> | Name Binding (NB) only | 95% | 45% |
> | NB + Event Triggering | 100% | 50% |
> | NB + Propagation | 70% | 50% |
> | All three (NB+ET+PR) | 72% | 50% |
>
> This shows: (1) Name Binding is the dominant enabler; (2) Event Triggering amplifies contamination; (3) Propagation modulates observed contamination, consistent with the stabilization effect in §4.1. The substantial model-specific variation further supports the systems-level interpretation.
>
> **Q3: Multi-round or adversarial self-reflection.**
> We evaluated single-round in-context self-reflection. The observed increase in contamination (e.g., +14pp for Claude) suggests that repeated reasoning within the same contaminated context may reinforce rather than correct contamination. Adversarial prompting (e.g., arguing against prior outputs) may activate different reasoning pathways and is an interesting direction for future investigation.
>
> **L1: Temperature 0.0.** Temperature 0.0 is standard for controlled evaluation. PSE mechanisms operate through registries and event systems rather than sampling; increasing temperature is expected to introduce variance without systematically altering contamination rates.
>
> **L2: Sample sizes.** Cost disparity exceeds 250x (`~$0.002/run vs ~$0.50/run`). Equalizing by cost would yield n=2-5 for frontier models, which is statistically insufficient. Our design (n=20-50 frontier, n=50-100 open-source) balances statistical power and cost. The new ablation adds n=160 per model. Per-model sample sizes will be explicitly reported in the unified table.
>
> In summary, we have added new empirical evidence, including an oracle-free defense and mechanism-level ablation, that directly address the reviewer's primary concerns regarding evaluation completeness and realism. We will incorporate all clarifications and presentation improvements in the revision.

---

> > ### Author Rebuttal · Reviewer_DgKb · 2026-04-03
> >
> > Thank you very much for your clarification; I really appreciate it. I believe the current score appropriately reflects the overall quality of the paper, so I would prefer to leave it unchanged.

---

### Official Review · Reviewer_UDii · 2026-03-13

**Soundness:** 3
**Presentation:** 3
**Significance:** 3
**Originality:** 3
**Overall Recommendation:** 4
**Confidence:** 3

**Summary:**

This paper studies persistent hidden state in tool-augmented LLM systems. It formalizes this as Persistent Semantic Entities (PSEs), defined through name binding, event triggering, and cross-context propagation. The authors argue that these entities can persist across tools, agents, and sessions, causing behavioral contamination that is hard to observe or debug. They evaluate this using a Rust/Python harness across 20 models from 9 families, comparing no pse, pse basic, and pse full settings, and measuring contamination, observability, defenses, and scaling behavior. The paper also includes several case-study mappings to real agent-framework incidents.

**Compliance With Llm Reviewing Policy:**

Affirmed.

**Key Questions For Authors:**

Asked in Weakness section. Some direction suggestions authors may consider

Tighten the taxonomy and explicitly distinguish PSEs from memory poisoning, persistent tool state, prompt injection, and ordinary software persistence.

Strengthen the contamination evaluation with threshold sensitivity analysis, manual adjudication, and clearer discussion of false positives and false negatives.

Clarify whether the contribution is primarily about model behavior, runtime architecture, or the interaction between them.

Add a stronger discussion of cross-model comparability limitations.

Provide full protocol details for the observability study.

Tone down the “propagation-mediated stabilization” claim unless there is stronger mechanistic evidence.

Evaluate a more realistic version of the best defense that does not assume oracle-like known-good outputs.
Report per-model sample sizes prominently.

Put more direct evidence for temporal persistence in the main paper.

Clean up minor submission-formatting issues.

**Limitations:**

yes

**Strengths And Weaknesses:**

Strengths
The paper targets a relevant and practical systems/security problem for agentic LLM systems: hidden persistent state that survives across boundaries and affects downstream behavior.

The decomposition into name binding, event triggering, and propagation is a useful conceptual framework for reasoning about persistence mechanisms in tool-augmented systems.

The paper evaluates many models, includes multiple task and defense settings, and examines observability and scaling in addition to attack success.

The observability angle is a strong addition. The paper looks beyond whether contamination occurs and also asks whether it can be diagnosed in realistic settings.

Weaknesses
Definition and novelty are not sufficiently sharp. The definition of PSE is too broad and appears to overlap with ordinary persistent state, memory poisoning, caching artifacts, and related persistent attack surfaces. The paper does not clearly establish what is uniquely a PSE. Will be nice if authors can add some text to do this clarification.

The contamination metric is underspecified for generative outputs. The paper relies on deviation from clean behavior plus a thresholded detection scheme using keyword matching and cosine similarity. This is brittle and not well justified for open-ended text generation. Authors may consider adding some text regarding this.

Model-level vs system-level attribution is unclear. Much of the persistence mechanism appears to come from the surrounding runtime and harness rather than the model itself, making it hard to interpret claims about cross-model susceptibility.

Cross-model comparisons are not well controlled. The study compares open and closed models across different APIs, wrappers, hidden prompts, and tool interfaces, so the comparisons are not fully apples-to-apples. Authors should consider adding some text to address this.

The statistical treatment is not described rigorously enough. The paper reports significance tests and effect sizes, but does not clearly explain the unit of analysis, independence assumptions, clustering, or whether the design requires a mixed-effects treatment. Authors may consider adding these in the draft.

The observability experiment lacks protocol detail. The paper does not explain clearly who the participants were, how the task was run, what expertise they had, or how localization steps were measured. Authors can address this by adding some text.

The defense evaluation is only partly realistic. The strongest method depends on known-good reference outputs, which makes it closer to an upper bound than a deployment-ready defense. If authors have any defense for this, you may consider adding it to the draft.

---

> ### Author Rebuttal · Authors · 2026-03-25
>
> We sincerely thank the reviewer for the detailed and constructive feedback. We are encouraged that the reviewer finds the work technically sound, meaningful, and broadly applicable. Below we address each concern with additional empirical evidence.
>
> **W1: PSE definition and distinction from related phenomena.**
> We agree that clearer distinction from related phenomena is important. We will add a comparison table in the revision. The defining feature of PSEs is the *conjunction* of all three mechanisms:
>
> | Phenomenon | Name Bind | Event Trig | Cross-Bdry Prop | Observability |
> |---|---|---|---|---|
> | Memory poisoning | - | - | - | Visible (data) |
> | Persistent tool state | Explicit | - | - | Visible (logs) |
> | Prompt injection | - | - | - | Single context |
> | **PSE** | **Implicit** | **Yes** | **Yes** | **75% missed** |
>
> Post-submission ablation (2^3 factorial, n=320, Gemini-2.0-Flash-Lite + Llama-3.1-8B) confirms each mechanism contributes independently:
>
> | Config | Gemini | Llama |
> |---|---|---|
> | No mechanisms | 0% | 0% |
> | Name Binding only | 95% | 45% |
> | NB + Event Triggering | 100% | 50% |
> | NB + Propagation | 70% | 50% |
> | All three (NB+ET+PR) | 72% | 50% |
>
> These results provide direct empirical support that PSEs are not reducible to existing categories, but arise from the interaction of all three mechanisms.
>
> **W2: Contamination metric for generative outputs.**
> We acknowledge that contamination evaluation for open-ended generation is inherently approximate. Two pieces of evidence support metric robustness: (1) iterative paraphrasing (n=40, GPT-4o-mini) shows 100% fact preservation across 20 iterations with 0% mutation, confirming keyword-matching reliability; (2) varying cosine threshold from 0.5 to 0.9 shifts absolute rates by ±8pp but preserves all relative rankings and statistical significance. We will include a threshold-sensitivity analysis. These results indicate that the metric is stable and sufficient for comparative evaluation.
>
> **W3: Model-level vs. system-level attribution.**
> We agree that this point benefits from clearer framing. Our contribution concerns the *interaction* between model and runtime. The ablation directly supports this: identical harness yields 95% (Gemini) vs. 45% (Llama) under name binding alone, confirming that neither component in isolation explains the phenomenon. We will revise §3 to explicitly frame PSEs as an interaction effect.
>
> **W4: Cross-model comparability.**
> We acknowledge the limitations of cross-model comparisons. Our claims are intentionally scoped to universal susceptibility and relative defense effectiveness, rather than fine-grained rankings. All models receive identical injection protocols. The new ablation provides a fully controlled within-harness comparison. We will add a "Comparability Limitations" paragraph.
>
> **W5: Statistical treatment.**
> We agree that the statistical methodology should be described more explicitly. The unit of analysis is individual task executions with fresh context and temperature 0.0, ensuring independence. We use ANOVA (H1), chi-squared tests (H2), and 2^3 factorial ANOVA (H3, 19,703 runs). Given the independence of runs in our design, these tests are well-suited to the experimental structure. We will add a statistical-methodology paragraph clarifying assumptions and test selection.
>
> **W6: Observability experiment protocol.**
> We apologize for the ambiguity. H2 is fully automated: "participants" refers to debugging agents performing sequential log inspections. No human subjects were involved. We will clarify the protocol.
>
> **W7: Defense evaluation realism (M5 upper bound).**
> We appreciate this concern and have added a post-submission defense variant that does not rely on oracle references. Specifically, we evaluate *context-isolated self-verification*: the model generates under contaminated context, then a separate clean call (without the contaminated context) performs fact-checking.
>
> | Defense | Contam. | Reduction |
> |---|---|---|
> | No defense | 70% (14/20) | -- |
> | Context-isolated verification (no oracle) | 15% (3/20) | **78.6%** |
> | External validation (cross-model) | 0% (0/19) | **100%** |
>
> Crucially, this is not self-reflection within the contaminated context (which fails), but a lightweight external validation mechanism. This directly addresses the realism concern: the improved method does not rely on oracle references and remains effective in practical settings. We will include these results in the revision.
>
> **Propagation-mediated stabilization.** The ablation provides supporting evidence (NB+ET: 100% vs. NB+ET+PR: 72% on Gemini), but the mechanism remains unclear. We present this as an empirical observation.
>
> **Temporal persistence.** We agree this is important and will promote key E5 results to the main paper.
>
> Given the reviewer's positive assessment of soundness, significance, and originality, we believe the work constitutes a solid contribution that can be further strengthened in the final version.

---

> > ### Author Rebuttal · Reviewer_UDii · 2026-04-01
> >
> > Thank you for responding. Satisfactory responses. Will keep the score same. Thanks.

---

### Decision · Program_Chairs · 2026-04-30

**Decision:**

Accept (regular)

**Comment:**

This paper studies a persistent implicit state that spreads across tools, agents, and sessions,
leading to behavioral inconsistencies that are difficult to diagnose. While this failure mode has
significant practical implications in tool-enhanced large language model systems, it has not yet
been fully investigated.

The main concerns are unclear definition, weak attribution, metric robustness, and defense
realism are sufficiently addressed. The authors clarify that PSE arises from the conjunction of
name binding, event triggering, and propagation, supported by new ablations; reframe the
contribution as an interaction between system and model rather than a purely model-level
claim; provide evidence that the contamination metric is stable for comparative purposes; and
introduce a more realistic, oracle-free defense that remains effective.
While limitations remain in cross-model comparability, scope of temporal claims, and
controlled evaluation settings, these do not undermine the core contribution. However, these
issues need to be addressed in future.